# MESHTRON: HIGH-FIDELITY, ARTIST-LIKE 3D MESH GENERATION AT SCALE

## ABSTRACT

Meshes are fundamental representations of 3D surfaces. However, creating high-quality meshes is a labor-intensive task that requires significant time and expertise in 3D modeling. While a delicate object often requires over $10^4$ faces to be accurately modeled, recent attempts at generating artist-like meshes are limited to 1.6K faces and heavy discretization of vertex coordinates. Hence, scaling both the maximum face count and vertex coordinate resolution is crucial to producing high-quality meshes of realistic, complex 3D objects. We present MESHTRON, a novel autoregressive mesh generation model able to generate meshes with up to 64K faces at 1024-level coordinate resolution –over an order of magnitude higher face count and $8\times$ higher coordinate resolution than current state-of-the-art methods. MESHTRON's scalability is driven by four key components: (*i*) an hourglass neural architecture, (*ii*) truncated sequence training, (*iii*) sliding window inference, and (*iv*) a robust sampling strategy that enforces the order of mesh sequences. This results in over $50\%$ less training memory, $2.5\times$ faster throughput, and better consistency than existing works. MESHTRON generates meshes of detailed, complex 3D objects at unprecedented levels of resolution and fidelity, closely resembling those created by professional artists, and opening the door to more realistic generation of detailed 3D assets for animation, gaming, and virtual environments.[1]

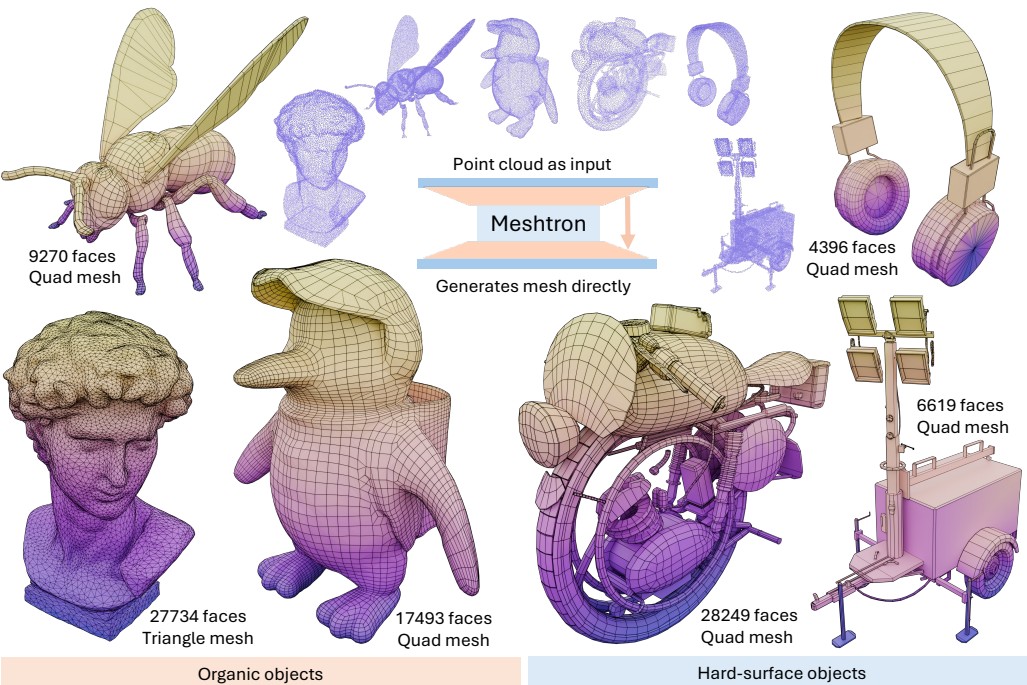

Figure 1: MESHTRON efficiently generates artist-style triangle or quad meshes of up to 64K triangle faces from point clouds. It sequentially generates mesh faces from bottom to top as illustrated by the color gradient. There are options to control the mesh density and produce quad-like topology.

---

[1]Anonymous project webpage: **https://meshtron.github.io/**

## 1 INTRODUCTION

Meshes are one of the most important and widely used representations for 3D assets. They are the de facto standard in the film, design, and gaming industries and are natively supported by virtually all 3D softwares and graphics hardwares. Currently, meshes can be generated either manually by artists or automatically through 3D generation algorithms. However, high-quality meshes suitable for games and movies –that is, those that can be manipulated, articulated, and animated– are still almost exclusively created by artists. Artist-created meshes capture not only the external appearance of objects but also their intrinsic properties and construction details through the *mesh topology* –the arrangement of vertices and faces in a specific tesselation. For instance, symmetrical objects should have a tesselation that mirrors their symmetry along the symmetry plane, and the edge flow should follow the natural curvature of the object to ensure smooth transitions and facilitate editing.

The availability of large-scale datasets and scalable models has driven remarkable advancements in generative models, such as autoregressive Large Language Models (LLMs) (Achiam et al., 2023; Team et al., 2023; Dubey et al., 2024), up to the point where generated data is often indistinguishable from real examples. Generating 3D assets directly as meshes, however, remains a significant challenge due to several factors: (*i*) the complexity of representing meshes properly due to their unordered, discrete nature, (*ii*) the large size of resulting representations, and (*iii*) the scarcity of high-quality training data. As a result, most of the 3D generative models avoid modeling meshes directly, and instead rely on alternative 3D representations such as neural fields (Poole et al., 2022; Wang et al., 2024), 3D Gaussians (Tang et al., 2024; Xu et al., 2024b), voxels (Brock et al., 2016; Wu et al., 2016), or point clouds (Vahdat et al., 2022; Jun & Nichol, 2023). These representations are then converted to meshes for downstream applications, for instance by iso-surfacing methods such as Marching Cubes (Lorensen & Cline, 1998; Chen & Zhang, 2021; Shen et al., 2023; Wei et al., 2023) and Marching Tetrahedra (Shen et al., 2021). Unfortunately, the resulting meshes often exhibit poor topology, characterized by overly dense tesselation, over-smoothing, and bumpy artifacts, leaving a significant quality gap between AI-generated meshes and those crafted by artists (Fig. 2).

To close this gap, recent works propose generating 3D as meshes by modeling the vertices and faces of the mesh directly, thereby avoiding the need for post-generation mesh conversion (Nash et al., 2020; Alliegro et al., 2023; Siddiqui et al., 2024; Weng et al., 2024; Chen et al., 2024a;b;c). While these approaches show promise for low face-count meshes, they are constrained by the extreme lengths of the resulting sequences. Consequently, they are limited to generating meshes up to 800 faces, with the sole exception of Chen et al. (2024c), which is capped at 1.6K faces. As a result, they fail to generate detailed, realistic shapes, which often consist of more than 10K faces. Figure 3 shows the face count distribution of our curated artist mesh dataset, which roughly follows a log-normal pattern with a mean of 32K faces and a median of 10K faces. As shown, existing methods are incapable of capturing the majority of artist-created meshes –indicated by the green and yellow vertical lines in Fig. 3– emphasizing the need for more scalable mesh generation methods.

Scaling up mesh generation to large, realistic meshes is a challenging task. The predominant way to represent a 3D mesh with N faces is by flattening into a *mesh sequence* of $3nN$ coordinates, where $n$ is the number of vertices per face. For a triangle mesh with 32K faces, this results in a sequence of 288K tokens. Generating sequences of this length presents challenges in terms of both efficiency and robustness. To address this limitation, recent works focus on developing compact representations that require less tokens to represent the mesh. For example, Siddiqui et al. (2024) uses a VQ-VAE to shorten the sequence by 33%, and Chen et al. (2024c) uses a lossless mesh compression algorithm to achieve about 50% reduction. Nevertheless, these reductions are still insufficient to close the large gap required to generate high-quality meshes of realistic objects.

In this paper, we take an orthogonal route. We focus on designing a more scalable and robust mesh generator by addressing the root of the scalability problem: the quadratic cost of vanilla Transformers. Existing methods rely heavily on global self-attention, making them prohibitively expensive when facing long mesh sequences required for detailed objects. We address this limitation through four key components: First, we recognize that mesh sequences have more structure than just being a homogeneous sequence of tokens; they can be equivalently represented at coordinate, vertex and face levels of abstraction. Additionally, emerging from the way mesh sequences are constructed, we identify a periodic pattern where ending coordinates within each face and vertex are harder to predict than starting ones (Fig. 5). Based on these insights, we replace the standard Transformer with a hierarchical *Hourglass Transformer* (Nawrot et al., 2021). This architecture introduces a strong

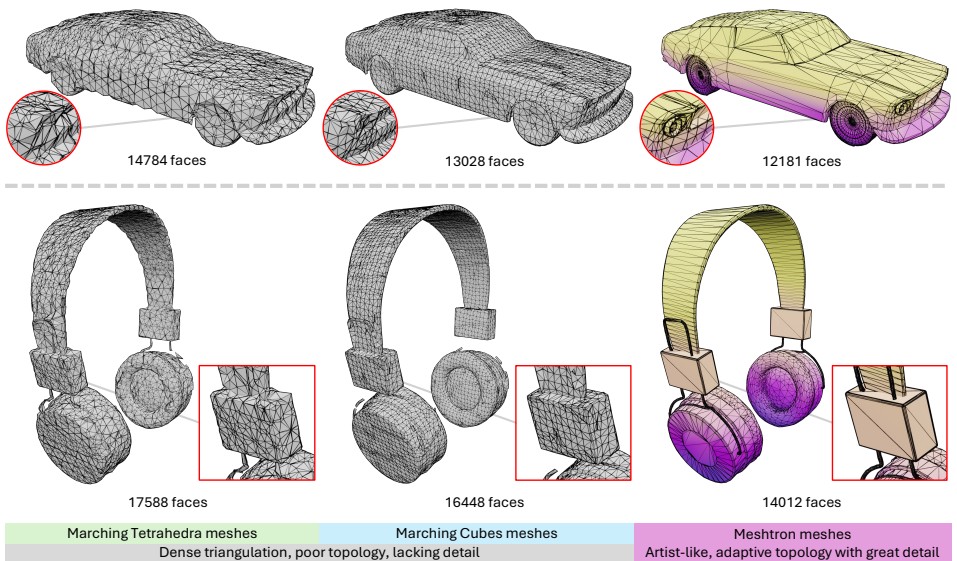

Figure 2: Topology comparison of MESHTRON and iso-surfacing methods DMTet (Shen et al., 2021) and FlexiCubes (Shen et al., 2023). While iso-surfacing methods can produce meshes with high face counts, they often suffer from overly dense tesselation, bumpy artifacts, oversmoothing and insufficient geometric detail, making them noticeably different from artist-created meshes. In contrast, MESHTRON produces meshes with high-quality topology, featuring high-geometric detail and well-structured tesselation that closely aligns with the standards of artist-created meshes.

inductive bias for mesh sequences by summarizing information hierarchically at increasing levels of abstraction aligned with vertices and faces. Additionally, the Hourglass architecture processes the hard-to-generate, last tokens of each vertex and face groups through deeper layers, efficiently allocating computational resources (Fig. 5b). Second, we observe that with proper conditioning, mesh generation does not require access to the full mesh sequence during training. Instead, we can train on truncated mesh sequences and generate complete mesh sequences during inference using a sliding window approach. This significantly reduces computation and memory costs during training, while also accelerating inference. Finally, we introduce a robust sampling strategy where subsequent coordinates must adhere to the predefined order in mesh sequences. This guarantees that generated mesh sequences maintain a realistic structure, leading to more consistent and reliable mesh generation.

Our proposed model, MESHTRON (Fig. 4), achieves over 50% reduction in training memory usage and $2.5\times$ higher throughput than existing methods. This efficiency allows us to train a 1.1B parameter autoregressive model capable of generating meshes with up to 64K faces and 1024-level vertex coordinate resolution using a simple distributed data parallel (DDP) setup. MESHTRON demonstrates unprecedented capability in generating artist-quality meshes, featuring detailed geometry, high-quality topology, and a high degree of diversity (Fig. 1,2). MESHTRON is conditioned on point-clouds and provides control over mesh density and quad-dominant generation, made possible through additional conditioning inputs introduced during training.

In summary, our contributions are:

- We uncover the periodical structure of a mesh sequence and exploit it to design an autoregressive model with better inductive biases based on Hourglass Transformer (Nawrot et al., 2021).

- We find that, with proper conditioning, generative mesh models can train on truncated mesh sequences and generate whole meshes with a sliding window strategy without performance loss.

- We provide a robust sampling strategy that guarantees generated mesh sequences to maintain a realistic structure, leading to consistent and reliable generation of very long mesh sequences.

- Based on these findings, we propose MESHTRON, a mesh generation model capable of generating meshes up to 64K faces at 1024-level vertex resolution, which is $40\times$ more faces and $8\times$ higher precision than existing works, while being faster despite it being $3\times$ larger (1.1B vs 350M). MESHTRON exhibits unprecedented mesh generation capabilities, setting a new benchmark and marking a substantial leap towards high-quality generation of 3D assets.

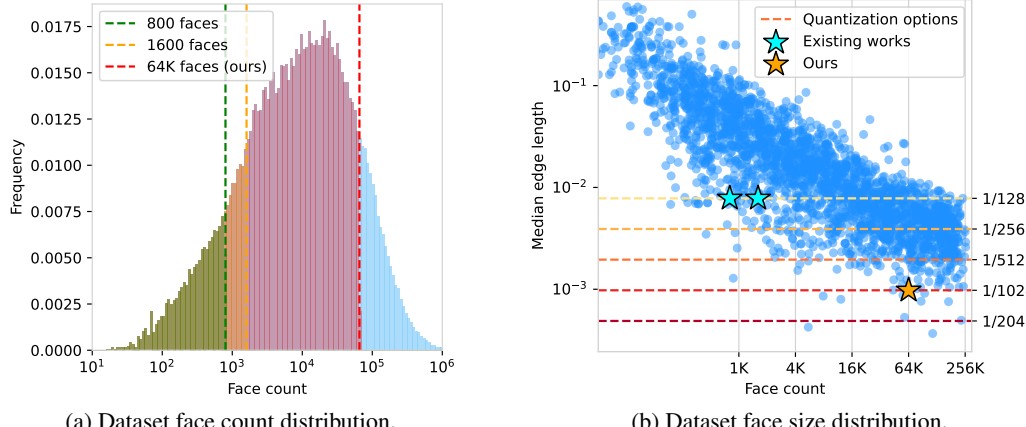

(a) Dataset face count distribution.  (b) Dataset face size distribution.

Figure 3: Distribution of face count (a) and face size (b) in a dataset of 1M artist-crafted meshes. The average face count is 32K, an order of magnitude higher than what current methods can generate. Moreover, meshes with higher face counts tend to have smaller faces. To accurately capture these details, the 128-level vertex quantization used in prior works must be increased –here to 1024 levels.

## 2 PRELIMINARIES – AUTOREGRESSIVE MESH GENERATION

**Meshes as ordered sequences of face, vertices and coordinates.** In this work, we formulate mesh generation as a sequence generation problem. Let $[n]$ denote the set $(1, 2, .., n)$. A mesh $\mathbf{M}$ of N faces $\{\mathbf{f}^i\}_{i \in [N]}$ is defined as the set of its faces: $\mathbf{M} = \{\mathbf{f}^1, \mathbf{f}^2, ...\mathbf{f}^N\}$, where each face $\mathbf{f}^i$ is an $n$-gon defined by $n$ vertices: $\mathbf{f}^i = \{\mathbf{v}_j^i\}_{j \in [n]}$. In the case of triangles, $n=3$, each face is defined by three vertices $\mathbf{f}^i = (\mathbf{v}_1^i, \mathbf{v}_2^i, \mathbf{v}_3^i)$, and each vertex $\mathbf{v}_j^i$ is represented by its coordinates: $\mathbf{v}_j^i = (\mathrm{v}_j^i x, \mathrm{v}_j^i y, \mathrm{v}_j^i z)$. Hence, a mesh can be described as a set of faces, vertices or coordinates of lengtha $N, 3N$ and $3n$N:

$$
\begin{aligned}
\mathbf{M} &= \{\mathbf{f}^1, \mathbf{f}^2, ... \mathbf{f}^N\} && \texttt{Face level} \\
&= \{\mathbf{v}_1^1, \mathbf{v}_2^1, \mathbf{v}_3^1, \ \mathbf{v}_1^2, \mathbf{v}_2^2, \mathbf{v}_3^2, \ ..., \mathbf{v}_1^N, \mathbf{v}_2^N, \mathbf{v}_3^N\} && \texttt{Vertex level} \quad (1) \\
&= \{\mathrm{v}_1^1 x, \mathrm{v}_1^1 y, \mathrm{v}_1^1 z, \ \mathrm{v}_2^1 x, \mathrm{v}_2^1 y, \mathrm{v}_2^1 z, ..., \mathrm{v}_3^N x, \mathrm{v}_3^N y, \mathrm{v}_3^N z, \ \mathrm{v}_3^N x, \mathrm{v}_3^N y, \mathrm{v}_3^N z\} && \texttt{Coord. level}
\end{aligned}
$$

**Autoregressive mesh generation.** In autoregressive mesh generation, a mesh $\mathbf{M}$ is generated by sequentially predicting each coordinate $c_i$ based on its conditional probability given all previously generated coordinates $\mathbf{c}_{<i}$: $p(c_i|\mathbf{c}_{<i})$. Then, the probability of the entire mesh is represented by the joint probability of all its coordinates:

$$
p(\mathbf{M}) = \prod_{i \in 3n\mathrm{N}} p\left(c_i|\mathbf{c}_{<i}\right). \quad (2)
$$

**From unordered to ordered mesh sequences.** For an autoregressive model to function properly, a consistent convention to order mesh sequences is required. Here, we adopt the convention introduced by Nash et al. (2020). First, vertices are arranged in $yzx$ order, where $y$ represents the vertical axis. Next, vertices within each face are sorted lexicographically, placing the lowest $yzx$-ordered vertex first. Finally, faces are sorted in ascending $yzx$-order based on the sorted values of their vertices. The resulting order can be seen through the color coding of the generated meshes presented in Figure 1.

In addition to vertex coordinates, we use three special tokens: start-of-sequence (S), end-of-sequence (E) and padding (P). We prepend 9 (S) tokens at the beginning of each mesh sequence and append 9 (E) tokens at the end. Padding tokens are used to fill batched sequences of different lengths. We always use these special tokens in groups of 9 to ensure that the structure of the mesh sequence is preserved at face and vertex levels in order to preserve structure in our Hourglass model (Sec. 3.1).

**Quantization of coordinates.** In autoregressive generation, models typically sample from a multinomial distribution over a discrete set of possible values. To follow this convention, we quantize the vertex coordinates $c_i$ into a fixed number of discrete bins. The resolution of the quantization grid, determined by the number of bins, directly affects the precision of the generated meshes. Higher quantization levels provide more detailed and accurate representations but increase the complexity of the generation process. Based on our analyses (Fig. 3b), we adopt a 1024-level quantization to accurately represent complex detailed meshes.

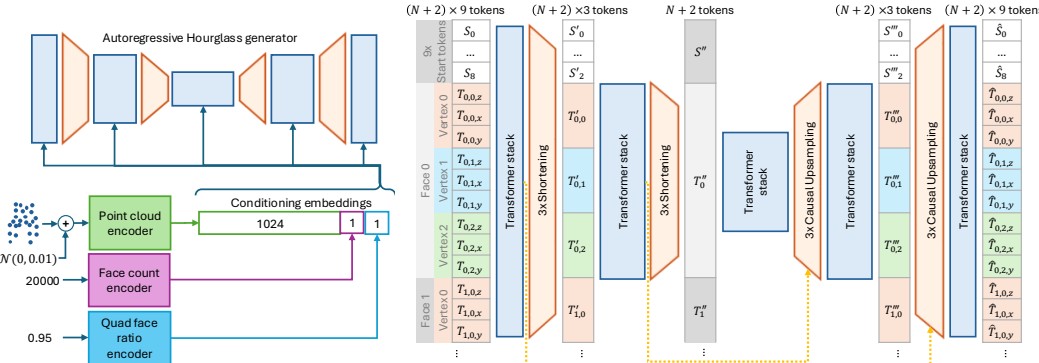

Figure 4: MESHTRON uses an Hourglass Transformer backbone with two shortening stages of factor $3\times$, conditioned on point-cloud, face-count and quad face ratio. Latent tokens are color-coded to show their relationship with the mesh sequence. Tokens at each shortened stage align with the vertices and faces of the mesh sequence, providing good inductive bias for mesh modeling.

## 3 MESHTRON– HIGH-FIDELITY MESH GENERATION AT SCALE

**Scaling up autoregressive mesh generation.** In autoregressive generation, a causal neural network –typically a Transformer (Vaswani, 2017; Achiam et al., 2023; Dubey et al., 2024)– is used to learn the conditional distribution $p(c_i|c_{<i})$. However, due to their quadratic computing complexity and linear memory requirements, handling long sequences quickly becomes prohibitively expensive. We address this limitation through the components detailed in the following subsections.

### 3.1 HIERARCHICAL MESH MODELING WITH HOURGLASS TRANSFORMERS

While mesh generation can be treated as a generic sequence generation problem, doing so overlooks the inherent structure of mesh sequences, which can be leveraged to build more effective models. As shown in Eq. 1, mesh tokens follow a two-level hierarchy where every 3 tokens represent a vertex, and every three vertices, i.e., 9 tokens, form a triangle. Unlike text tokens, individual mesh tokens carry limited information and become meaningful only when processed in the respective hierarchical groups. By treating mesh tokens as cohesive hierarchical units, models could better capture the structure of mesh sequences, leading to improved understanding and generation. In addition to the hierarchical structure of mesh sequences, we observed a distinctive repetitive pattern in the difficulty of generating tokens resulting from the ordering used to construct mesh sequences (Sec. 2). As shown in Fig. 5, early tokens within each triangle are easier to generate, as evidenced by lower average perplexity values, than later ones. Similarly, within each vertex, later tokens tend to be harder to predict. This pattern arises from vertex sharing across adjacent triangles, which introduces repeated tokens in the mesh sequence (Fig. 5a).

These insights draw our attention to the *Hourglass Transformer* architecture (Nawrot et al., 2021), an autoregressive hierarchical model designed to process inputs at multiple levels of abstraction (Fig. 4). The architecture employs multiple Transformer stacks at each level, with transitions between levels managed by causality-preserving shortening and upsampling layers that bridge these hierarchical levels. Shortening layers compress groups of token embeddings into a single embedding via average, linear or attention-based pooling. Upsampling layers reverse this process by expanding a single embedding back into multiple tokens using repeating, linear upsampling or attention-based upsampling. The expanded sequence is then combined with the higher-resolution sequences from early levels via residual connections, similar to U-Nets (Ronneberger et al., 2015). Both shortening and upsampling layers are carefully designed to preserve causality, as detailed in Fig. 10. In addition, the Hourglass architecture offers a static routing mechanism that allocates compute differently to tokens based on their positions within the sequence. For a shortening factor of $s$, only every $s^{\text{th}}$ token in the sequence passes through the inner Transformer stacks, while other tokens bypass it. For instance, with $s=3$, every $3^{\text{rd}}$ token is processed through the full stack, while every $1^{\text{st}}$ and $2^{\text{nd}}$ token skip the inner stacks. This selective routing enables the model to distribute compute efficiently given a prespecified structure of the input. An animation illustrating this process is available **here**.

We design the backbone of MESHTRON as an Hourglass Transformer with two shortening layers, each reducing the sequence by a factor of 3. This results in a three-stage model, where the shortened stages correspond to token groups representing the vertices and faces of a mesh (Fig. 4). By align-

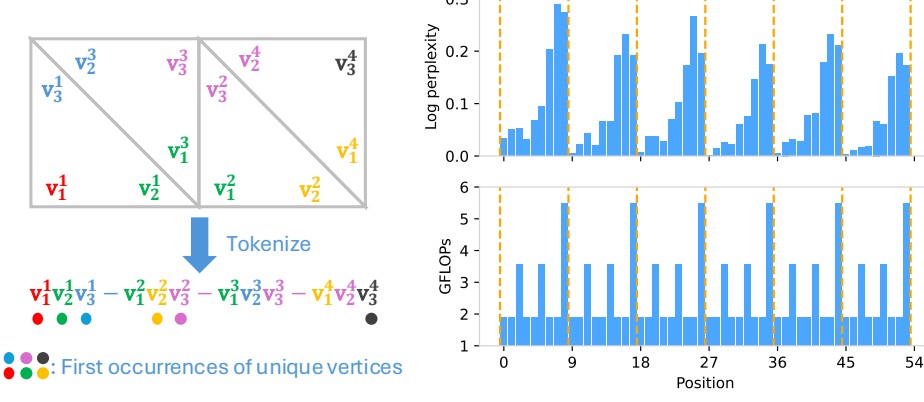

(a) Repeated vertices in a mesh sequence.  (b) Per-token log-perplexity and computing.

Figure 5: Not all mesh tokens are equal. (a) illustrates ordering of tokens in mesh sequences. (b) shows per-token log perplexity averaged over 1K mesh sequences (top) and the compute allocated per token by the Hourglass architecture. Groups of 9 tokens forming a triangle are marked with dashed vertical lines. Earlier tokens in a triangle show lower perplexity, as the first two vertices are often shared with previous triangles. The last vertex is less constrained, therefore introducing greater uncertainty. The Hourglass Transformer captures this periodicity and allocates more compute to high-perplexity token positions, making it more effective for mesh generation –see animation **here**.

ing the architecture with the structural patterns observed in (1) and Fig. 5, MESHTRON allocates resources more effectively than the vanilla architectures used in previous works. As shown in Sec. 4.1, MESHTRON leads to improved training and inference efficiency along with superior overall results.

## 3.2 TRAINING ON TRUNCATED SEQUENCES AND INFERENCE WITH SLIDING-WINDOW

Mesh sequences can be extremely long, ranging from a few to hundreds of thousands of tokens (Fig. 3a). As a result, training on full mesh sequences can still be prohibitively expensive, even with the memory and computation savings of an Hourglass architecture. Moreover, the wide variation in sequence lengths hinders the implementation of efficient training setups, even with advanced parallelization techniques in place (Liu et al., 2023a; Korthikanti et al., 2023).

Fortunately, mesh generation has a special property that can be exploited to scale generation to very long mesh sequences. The ordering of the mesh sequence sorts triangles from bottom to top, layer by layer, promoting the locality of adjacent triangles within the sequence (Sec. 2). Hence, assuming proper global conditioning (see Sec. 3.3 for details on how to achieve this), the generation of subsequent triangles only requires information from adjacent tokens, –specifically, the vertex positions of nearby triangles. This special property allows us to adopt a sliding window approach for efficient training and inference ((Fig. 6 left). Specifically, we train our model with fixed-length truncated segments of mesh sequences to significantly reduce compute and memory consumption during training. Then, during inference, we use a rolling KV-cache with a buffer size equal to the attention window, to achieve linear complexity. Importantly, while there is a small discrepancy between training and inference due to cached embeddings carrying information from outside the current attention window during inference (Fig. 6 right), Sec. 4.2 shows that this has no negatively impact on performance, allowing for efficient generation without the need to recompute previous contexts.

## 3.3 GLOBAL CONDITIONING ON TRUNCATED SEQUENCES WITH CROSS-ATTENTION

Recent works, such as MeshXL (Chen et al., 2024a) and MeshAnything (Chen et al., 2024b;c) perform conditional generation by attaching the embedding of the conditional variables, e.g., point-clouds, at the beginning of mesh sequences. However, as our scaling strategy involves training on truncated mesh sequences, prepending would either (*i*) make the conditional signal visible to only a few mesh segments, or (*ii*) require complex concatenation strategies during training and inference. To overcome these limitations, we use *cross-attention* to condition all mesh segments on global conditioning signals, irrespective of their position within the sequence. This enables our model to effectively combine local and global information during both training and inference, resulting in accurate predictions while keeping low resource usage.

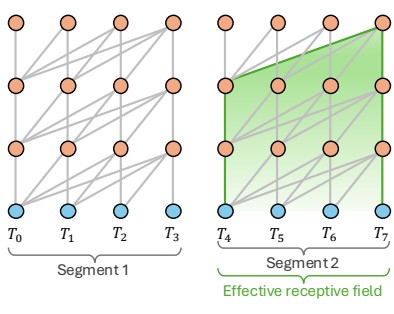 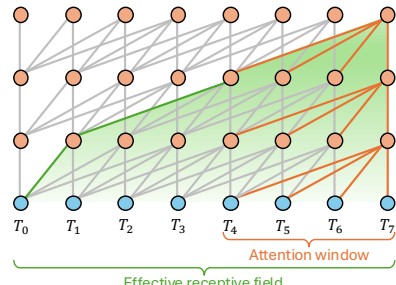

(a) Train on cropped segments.   (b) Inference with rolling KV-cache or sliding-window attention.

Figure 6: Extended receptive field during inference. While MESHTRON is trained on truncated segments, it employs a rolling KV-cache during inference for efficiency. This creates a mismatch between training and inference, as the effective receptive field increases during inference and cached latents can carry over information from far beyond the training receptive field. Interestingly, we find the overall effect to be beneficial compared to training on full mesh sequences (Sec. 4.2).

MESHTRON is designed for point-cloud conditioned mesh generation model. Point clouds are a flexible and universal 3D representation that can be efficiently derived from other 3D formats, including meshes. We encode the input point-cloud into 1024 embeddings via a jointly-trained Perceiver encoder (Jaegle et al., 2021; Zhao et al., 2024). Additionally, we condition generation on the face count and the proportion of quad faces in the mesh before triangulation. These variables allow control over mesh density and quad-dominance during inference. Each variable is encoded into 1 embedding via an MLP and concatenated to the point-cloud embeddings for conditioning (Fig. 4). Following Dubey et al. (2024), we replace every $4^{\text{th}}$ layer in the Transformer stack with a cross-attention layer to enable interaction between the main model and the conditioning embeddings.

### 3.4 ROBUST MESH GENERATION WITH MESH SEQUENCE ORDERING ENFORCEMENT

To ensure robust generation, we enforce generated mesh sequences to adhere to the ordering with which they are constructed (Sec. 2). Specifically, we constrain generation such that vertex coordinates within each face follow a lexicographic ascending order, and the coordinates of subsequent faces also follow a lexicographic ascending order relative to previous faces. Additionally, we constrain end-of-sequence tokens to appear only at the start of a new face. These constraints prevent the generation of inconsistent sequences, ensuring that the outputs remain within the data distribution.

We benchmark our order enforcement algorithm on the validation dataset by simulating the generation process. For each token, we calculate the number of invalid categories in the $N$-way categorical distribution that would violate the sequence ordering based on prior tokens. Our algorithm prevents 32% invalid predictions at 1024-level quantization and 27% at 128-level quantization, effectively narrowing the model's sample space and enhancing both generation quality and robustness.

## 4 EXPERIMENTS

This section is divided in two parts. First, we validate the components introduced in MESHTRON on a small-scale setup with models of 500M parameters and a dataset with meshes up to 4096 faces and 128-level coordinate quantization. Next, based on the insights from these experiments, we conduct a full-scale study using a model with 1.1B parameters and a dataset with meshes up to 64K faces and 1024-level coordinate quantization. During training, we employ strategies to enhance MESHTRON's generalization to imperfect, non-artist meshes. First, we avoid sampling points from internal mesh structures, often absent in non-artist meshes. Then, we perturb the point cloud with Gaussian noise, applying up to $\sigma_{\text{pos}}{=}0.1$ to the point positions, and $\sigma_{\text{normal}}{=}0.2$ to the point normals. This improves generalization and provides a mechanism to balance creativity and faithfulness by increasing noise or avoiding noise (Fig. 11). Further details on architectural configurations, augmentations, training setups, hyperparameters and datasets used are provided in Appx. C.1.

### 4.1 HOURGLASS VS PLAIN TRANSFORMER

First, we validate our choice of the Hourglass Transformer over a plain Transformer architecture. To this end, we compare a plain transformer and two Hourglass models with similar architectures but different layer distributions in their Transformer stacks. We report training memory usage, training speed, validation perplexity and the reconstruction accuracy of the generated meshes.

Table 1: Hourgrass vs. plain Tranformer. Plain-X depicts a plain Transformer with X blocks. HG-X-Y-Z depicts an hourglass with Hourglass X blocks at full resolution, Y blocks at 1/3 resolution, and Z blocks at 1/9 resolution. We report peak training memory at batch size 1 and 2, wall clock training time and actual inference speed in our non-optimized setting. We evaluate symmetric Chamfer distance, which has an oracle score of $0.986 \times 10^{-2}$ on the validation set.

| Architecture | Pos. Emb. | Train Iters. | Train Hours | Memory GB↓ | Inference Tok/s↑ | Val PPL↓ | Chamfer↓ ($\times 10^{-2}$) | Avg. # Faces |
|---|---|---|---|---|---|---|---|---|
| Plain-24 | LPE | 100K | 50 | 34.6 / 64.4 | 57.4 | 1.077 | 1.176 | 937 |
|  | RoPE | 100K | 50 |  |  | 1.074 | 1.105 | 889 |
| HG-8-8-8 | RoPE | 100K | 26 | 20.1 / 35.2 | 108.2 | 1.075 | 1.080 | 876 |
|  | RoPE | 190K | 50 |  |  | **1.066** | 1.083 | 873 |
| HG-4-8-12 | RoPE | 100K | 22 | **16.1 / 27.2** | **144.7** | 1.076 | 1.127 | 939 |
|  | RoPE | 230K | 50 |  |  | 1.067 | **1.044** | 953 |

Table 2: Performance of models trained on truncated sequences. The HG-8-8-8 architecture is used in this experiment. We increase the batch size for the truncated model to maintain a comparable token-per-batch. *Many sampling attempts failed to produce a stop token.

| Train Segments | Memory (GB) ↓ | Window Size | Val. PPL ↓ | Chamfer ($\times 10^{-2}$) ↓ | Avg. # Faces |
|---|---|---|---|---|---|
| 4096 (full) | 20.1 / 35.2 | – | 1.066 | 1.083 | 873 |
| 1024 | **8.9 / 12.8** | – | 1.221 | 2.212 | 1258* |
|  |  | 1024 | **1.059** | **1.016** | 808 |

As shown in Table 1, the Hourglass model HG-8-8-8 matches or surpasses the plain model under the same number of training iterations, while saving 40% of memory and being almost twice as fast both in training and inference. When trained for the same wall-clock duration, HG-8-8-8 significantly outperforms the plain Transformer. The lighter HG-4-8-12 model, while slightly less effective under the same iteration count, achieves the best reconstruction accuracy when trained for the same wall-clock duration. It is also $2.5\times$ faster and uses less than half the memory of a plain Transformer. These results confirm the superior efficiency and effectiveness of Hourglass for mesh generation. Additionally, we compare learnable positional embedding (LPE), commonly used in mesh generation, with rotary positional embeddings (RoPE) the de-facto standard in language modeling. RoPE delivers better performance and is natively compatible with the rolling KV-cache (Tab. 1).

### 4.2 GENERATING FULL MESH SEQUENCES WITH MODELS TRAINED ON TRUNCATED DATA

Next, we study the sequence length generalization capability of the Hourglass model by training it on truncated mesh sequences of up to 1024 faces and evaluating its performance generating full mesh sequences with and without sliding window attention (SWA). Surprisingly, as shown in Table 2, the model trained on truncated sequences outperforms the model trained on full mesh sequences, while reducing memory usage by over 50%. This result confirms that, with proper global conditioning, mesh generation does not require access to the entire mesh sequence. However, omitting SWA during inference significantly degrades performance, highlighting its importance in bridging the train-test gap. As shown in Fig. 7, validation perplexity degrades rapidly beyond the training length without SWA –a known phenomenon in the LLM community (Press et al., 2021; Sun et al., 2022).

### 4.3 SCALING MESHTRON TO 64K FACES

Our previous studies confirm that the Hourglass architecture offers better inductive biases for mesh generation and that, with proper global conditioning and sequence ordering, mesh generation can effectively rely on local information without sacrificing performance. With our design validated at a small scale, we now scale up MESHTRON to 1.1B parameters and a dataset containing meshes up to 64K faces with 1024-level coordinate quantization. Higher quantization is required to accommodate for the smaller face sizes encountered at this scale (Fig.3b) and to enhance geometric quality.

We evaluate MESHTRON as a retopologization tool based on meshes from three different sources: artist-created, 3D-scanned, and generated through iso-surfacing by online text-to-3D services. Point clouds sampled from these meshes are used to condition our model. We compare MESHTRON with artist-like mesh generators MeshAnythingV1/V2 (Chen et al., 2024b;c) –the state-of-the-art point cloud conditioned artist-like mesh generator– and MeshGPT (Siddiqui et al., 2024) (Figs. 8, 9, 14), as well as point-cloud conditioned iso-surfacing methods DMTet (Shen et al., 2021) and FlexiCubes (Shen et al., 2023) (Fig 2). An extended gallery of results is available on our **website**.

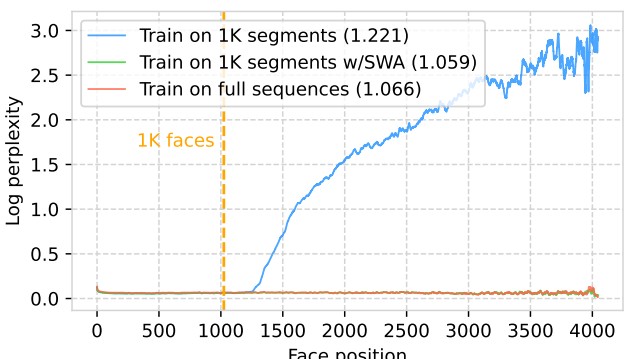

Figure 7: Context generalization of models trained on truncated mesh sequences. Naively extending the context window beyond the length of training leads to poor inference (blue). This is addressed by applying sliding-window attention (SWA) (green). For comparison, a model trained and inferred on full sequences is also shown (red). Mean validation perplexity shown in parenthesis (lower is better).

Thanks to its greatly expanded context length, MESHTRON handles complex shapes significantly better than MeshAnythingV2. Its use of $8\times$ higher resolution for vertex coordinate quantization also leads to smoother meshes. On out-of-distribution, non-artist meshes, MESHTRON faithfully reproduces input shapes, whereas MeshAnythingV2 struggles. This generalization and robustness in unseen noisy settings can be attributed to the training-time augmentations used. These results highlight MESHTRON's emerging ability as a remeshing tool, capable of improving the tessellation of existing meshes with poor topology –whether from scans or AI generation tools. Additionally, we validate the effectiveness of quad-mesh conditioning, where MESHTRON successfully generates triangulated quad meshes that can be converted into high-quality quad meshes using off-the-shelf algorithms. This demonstrates new opportunities for data-driven approaches to tackle the traditionally challenging task of quad remeshing. Unlike iso-surfacing methods (Fig. 2), MESHTRON produces meshes with high-quality topology, detailed geometric detail and well-structured tesselation.

## 5    LIMITATIONS AND FUTURE WORK

Despite the advancements offered by MESHTRON, several areas for improvement remain. First , even with MESHTRON's greatly improved efficiency, generating large meshes still requires considerable time –our largest model inferences at 140 tokens / sec. Advancements in efficient autoregressive models (Gu & Dao, 2023; Poli et al., 2023), speculative decoding (Cai et al., 2024; Leviathan et al., 2023) and advanced inference systems (Kwon et al., 2023), could help accelerate this process. Second, while MESHTRON exhibits impressive generation capabilities, it is limited by the low-level nature of its point cloud conditioning. As a result, it struggles to add significant detail to degraded 3D shapes, e.g., from marching cube text-to-3D generators. Incorporating higher-level conditioning signals such as text, or additional guidance, e.g., high-resolution surface normal maps, could further enhance its capabilities. Another challenge lies in ensuring the robustness of autoregressive generation for long sequences –an area that remains underexplored. Despite our sequence ordering enforcement, occasional failures still occur during inference, expressed by missing parts, holes, or non-termination. This challenge warrants further investigation. Lastly, the scarcity of high-quality 3D data may hinder the advancement of data-driven methods like MESHTRON. While MESHTRON is very scalable, it relies on a massive amount of 3D data to train. However, the availability of high-quality 3D data significantly lags behind other modalities. Leveraging data from other domains, e.g., images or videos, to support mesh generation is a promising direction to address this limitation.

## 6    CONCLUSION

We introduce MESHTRON, a point cloud-conditioned autoregressive mesh generation model capable of producing artist-quality meshes with up to 64K faces at 1024-level coordinate resolution. Leveraging the Hourglass architecture, MESHTRON captures the intrinsic structure of mesh sequences, improving efficiency and performance. By combining effective global conditioning with training on truncated mesh sequences, it efficiently utilizes both local and global information during generation. Controlled experiments validate its components, guiding an optimized scaling strategy. MESHTRON exhibits unprecedented capabilities, generating high-quality, artist-like meshes directly from point cloud inputs with fine control over density and tessellation style, significantly outperforming prior work in face count, spatial resolution and quality, enabling the generation of complex, realistic 3D meshes. MESHTRON marks a major step toward practical, artist-friendly mesh generation tools, automating retopologization and reducing the manual effort required for high-quality 3D modeling.

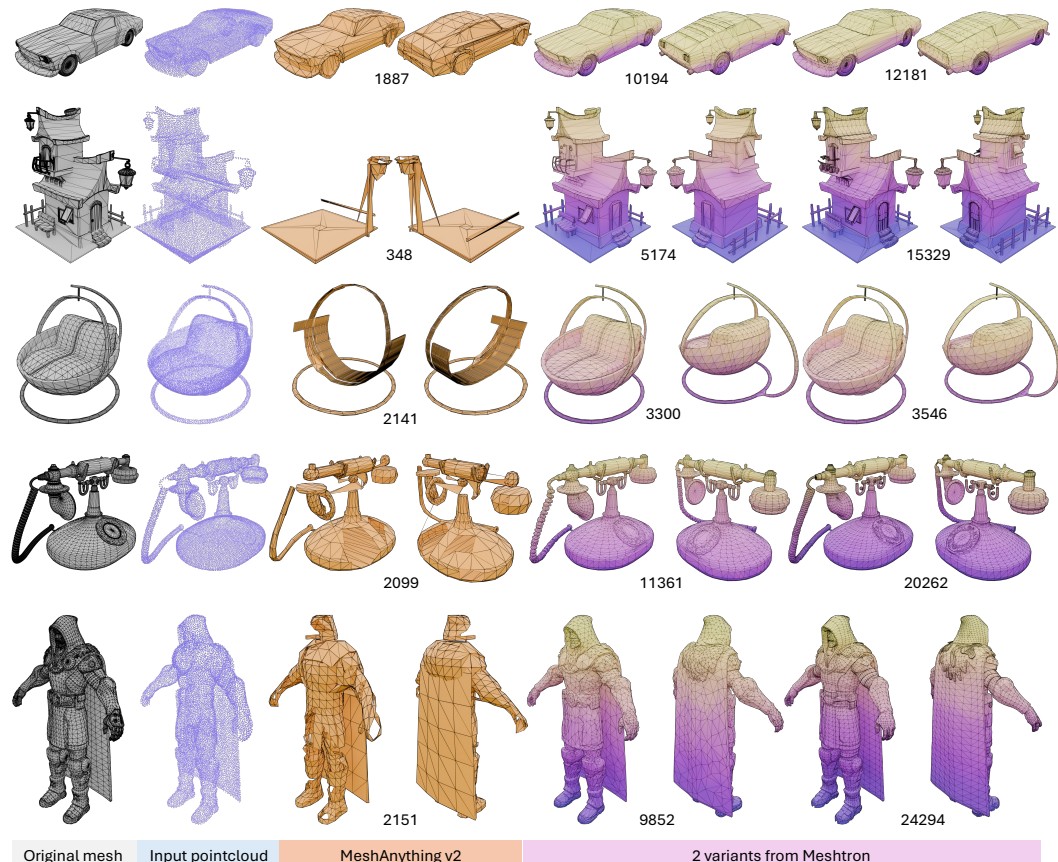

Figure 8: Comparison on point cloud conditioned mesh generation. Point clouds are sampled from existing artist mesh shown on the left. The face counts are noted below each mesh. For each shape, we provide 2 variants to demonstrate the generation diversity as well as the face density control capability of Meshtron.

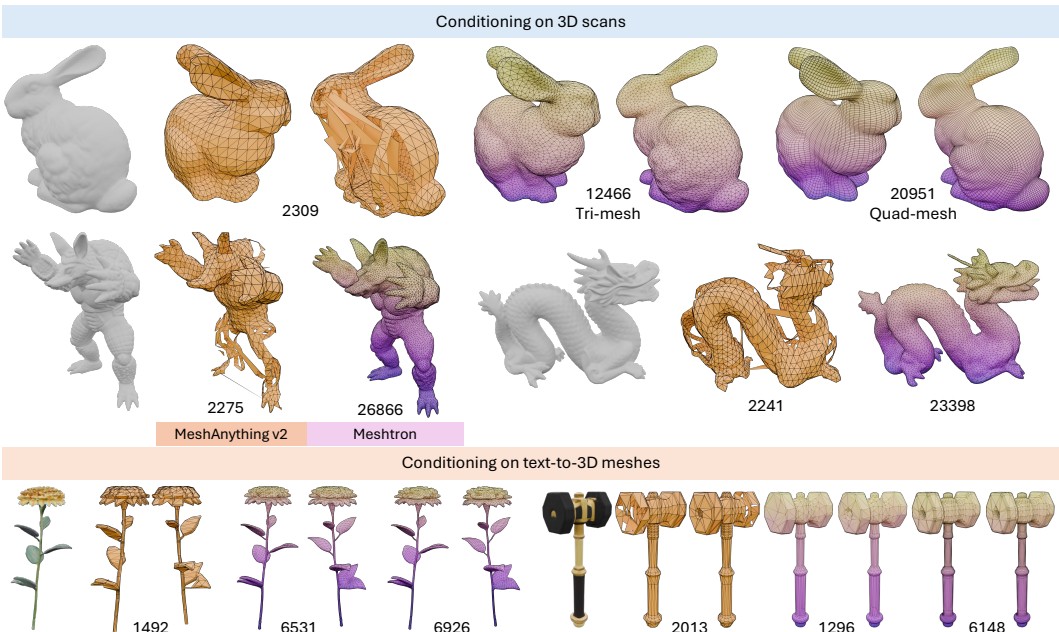

Figure 9: Comparison on conditional mesh generation with point clouds sampled from non-artist meshes. Meshes coming from 3D scanning or text-to-3D tools are usually very dense and having poor topology. Meshtron can be used as a remeshing tool to improve the tessellation of these meshes.

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

# APPENDIX

## A  EXTENDED RELATED WORK

**3D generation.** There is a plethora of work on 3D generation, where text or images are commonly used as inputs to generate corresponding 3D shapes. These works utilize a wide range of internal 3D representations. Poole et al. (2022); Wang et al. (2024); Liu et al. (2023b); Qian et al. (2023); Tang et al. (2023b); Shi et al. (2023); Li et al. (2023); Lorraine et al. (2023) represent the shapes using neural radiance fields (NeRF) (Mildenhall et al., 2021), enabling supervision through differentiable volumetric rendering. Gao et al. (2022); Lin et al. (2023); Liu et al. (2024); Chen et al. (2023); Jun & Nichol (2023); Qian et al. (2023); Long et al. (2024); Wei et al. (2024); Bensadoun et al. (2024) adopts signed distance fields (SDF), another popular 3D representation. Closely related to SDF is occupancy fields, which is used in Zhang et al. (2024); Peng et al. (2020). Finally, 3D Gaussian (Kerbl et al., 2023) is also gaining popularity in 3D generation literature (Tang et al., 2023a; Ren et al., 2023; Tang et al., 2024), bringing improved efficiency and quality.

**Mesh extraction and reconstruction.** The above-mentioned body of work on 3D generation often requires additional mesh extraction steps to convert the outputs into meshes suitable for downstream applications. The foundation of these methods are Marching Cubes (Lorensen & Cline, 1998) and the closely related Marching Tetrahedra (Doi & Koide, 1991) algorithms, each implementing a set of hand-designed rules to extract a mesh from the iso-surface of a 3D volume. Their modern derivatives (Shen et al., 2023; 2021; Chen & Zhang, 2021; Wei et al., 2023) are extensively utilized in the 3D generation literature to refine the geometry and extract the meshes.

**Data-driven mesh generation.** The iso-surfacing methods discussed above focus on modeling the geometry of the 3D representations and do not consider the resulting mesh tesselation during optimization. Consequently, resulting meshes are often overly tessellated, and do not resemble those created by artists (Fig. 2). Recent works have begun addressing this issue by learning the tessellation directly from artist-created meshes. These models often employ an autoregressive backbone: PolyGen (Nash et al., 2020) adopts two autoregressive models for generating the vertices and faces of a polygon mesh respectively. MeshGPT (Siddiqui et al., 2024) couples a VQ-VAE, which shortens the sequence length by $1/3$, with an autoregressive model. PivotMesh (Weng et al., 2024) introduces pivot vertices to improve generation quality. MeshXL (Chen et al., 2024a) scales up the model and data while using a single autoregressive model to generate raw mesh sequence directly. MeshAnythingV1 (Chen et al., 2024b) adds conditional generation capability to MeshGPT by including a point cloud encoder. MeshAnythingV2 (Chen et al., 2024c) extends support to larger meshes of up to 1.6K faces by replacing the VQ-VAE with a more efficient, lossless mesh compression algorithm.

There also exist works that use different generative formulations or mesh representations. BSP-Net (Chen et al., 2020) generates compact meshes via binary space partitioning. PolyDiff (Alliegro et al., 2023) generates meshes using a discrete diffusion model. BrepGen (Xu et al., 2024a) generates parametric surfaces instead of polygons using diffusion models. DeepCAD (Wu et al., 2021) generates CAD commands that constructs the shape instead of the mesh itself.

**Efficient Transformer models.** There has been pursuit for efficient long sequences support for autoregressive Transformer models driven mainly by the LLM community. Several works utilize sparse attention to reduce complexity (Child et al., 2019; Beltagy et al., 2020; Zaheer et al., 2020; Ding et al., 2023). Notably, Longformer (Beltagy et al., 2020) proposes using sliding-window attention to achieve linear computing cost with regard to the sequence length, which has been adopted in popular LLMs such as Mistral (Jiang et al., 2023), Gemma 2 (Team et al., 2024), and GPT-NEO (Black et al., 2021). As an under-explored approach that also reduces model complexity, Nawrot et al. (2021) proposes a hierarchical transformer architecture that considerably conserves memory and computing by reducing the sequence length using causal shortening and upsampling.

Apart from reducing model complexity, another promising avenue of reducing the training cost is to train on shorter sequences while still generating full sequences during inference. Training-free long-context methods such as An et al. (2024); Xiao et al. (2023) extend pretrained models beyond its training sequence length without finetuning. As for training-based methods, Dai et al. (2019) finds the positional embedding crucial for chunk-based training. Sun et al. (2022) discovers that damping the magnitude of RoPE positional embedding improves the extrapolation performance, and windowed attention is also beneficial by improving the "attention resolution".

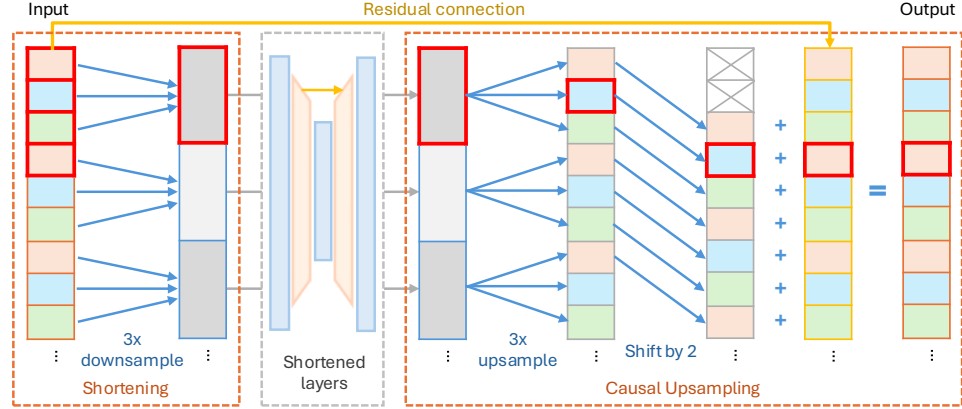

Figure 10: Causality preservation in Hourglass Transformers (Nawrot et al., 2021). Each level of the Hourglass hierarchy encompasses a shortening (downsampling) layer, upsampling layer, and the stack of layers in between that operate on shortened sequences. To preserve causality, for a shortening value $s$, the upsampled sequence is shifted by $s-1$, before being combined with the embeddings from the previous level via a residual connection ($s=3$ in our case). We illustrate the causal property of the Hourglass model by marking all embeddings contributing to one of the outputs with red boxes. We refer the reader to Nawrot et al. (2021) for further details.

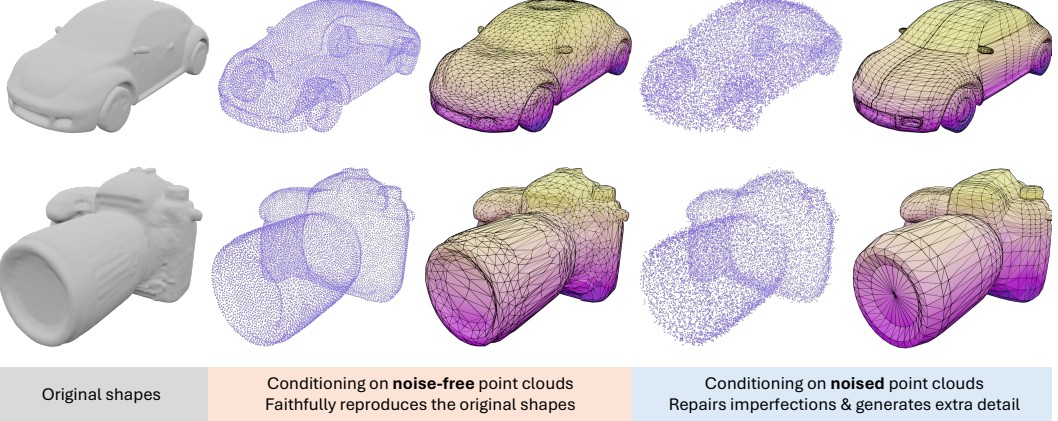

Figure 11: Balancing faithfulness and creativity in MESHTRON. Adjusting the noise level in the input conditioning point cloud allows MESHTRON to balance between full faithfulness, possibly preserving imperfections from scanned or iso-surfacing meshes, and creativity, allowing more flexible detail enhancement and topology refinement.

## B CAUSALITY PRESERVATION IN HOURGLASS TRANSFORMERS

The Hourglass architecture adopted by MESHTRON involves carefully designed shortening and up-sampling layers that preserve causality. By doing so, *information leak* is avoided (Fig. 10).

## C ADDITIONAL EXPERIMENTAL DETAILS

### C.1 MODEL ARCHITECTURE

Our model consists of two main components: a point cloud encoder and an autoregressive Hourglass Transformer. For the encoder, we use an 8-layer Transformer in the small-scale models and a 12-layer Transformer in the full-scale experiment. We use input point clouds with 8192 points for small-scale experiments and 16384 points for the full-scale experiment. For the small-scale Hourglass models, the number of layers and channels remain fixed, with shortening and upsampling layers inserted into the network. In the full-scale model, we adopt the HG-4-8-12 configuration. Further details can be found in Table C.1. All models are trained with a cosine scheduler from $1\mathrm{e}^{-4}$ to $1\mathrm{e}^{-5}$ using linear learning rate warm-up for 5K iterations and a weight decay of $1\mathrm{e}^{-2}$.

Table 3: MESHTRON's architectural and training details.

| | Small scale | Full scale |
|---|---|---|
| Architecture | Straight HG-8-8-8 HG-4-8-12 | HG-4-8-12 |
| Layers | 24 | 24 |
| Channels | 1024 | 1536 |
| Head channels | 64 | 96 |
| FFN hidden channels | 2816 | 4096 |
| Activation function | SwiGLU | SwiGLU |
| Cross attention interval | 4 | 4 |
| Shortening type (for Hourglass only) | Linear | Linear |
| Upsample type (for Hourglass only) | Linear | Linear |
| RoPE theta | 1M | 1M |
| Coord. quantization steps | 128 | 1024 |
| Point encoder layers | 8 | 12 |
| Point cloud size | 8192 | 16384 |
| Training chunk size | – | 8192 |
| Parameter count | 0.5B | 1.1B |

## C.2    DATA HANDLING

**Data curation.** Our training data is licensed from a major 3D content provider. We curate the data by reviewing rendered images to remove meshes with low geometric quality. Additionally, we remove all scanned, reconstructed and decimated meshes, as well as those produce by a CAD software, using a combination of metadata keyword filtering and geometric heuristics. This ensures that non-artist meshes do not compromise the quality of the trained model. The final dataset contains of 700K meshes containing fewer than 64K faces.

Many meshes in the dataset contains quad faces, with a small subset being consisting entirely of quads. During training, we triangulate these meshes but retain the original quad percentage and use it as conditioning. This allows for control over the generation of (triangulated) quad meshes.

**Training time augmentations.** During training, we apply data augmentations including random rotations, translations and scaling. For point cloud conditioning, we sample points from the mesh surface by rasterizing the mesh from 20 viewpoints corresponding to the 20 faces of an icosahedron. Then, depth maps are extracted from these views and unprojected onto point clouds to filter points coming from inner structures in the mesh. Next, we subsample the point cloud using farthest-point sampling. This approach avoids sampling points inside objects, improving MESHTRON's generalization to non-artist meshes, which often lack internal structure.

Additionally, we perturb the point cloud with Gaussian noise, applying of up to $\sigma_{\text{pos}}=0.1$ to the point positions, and $\sigma_{\text{normal}}=0.2$ to the point normals. We also randomly set the point normals to zero vectors for the whole point cloud with a chance of $0.5$. These perturbations enhance the model's generalization and provides a mechanism to balance creativity (by adding more noise) and faithfulness (by reducing or avoiding noise) during inference (Fig. 11).

## C.3    EVALUATION PROTOCOL

Since our models are always conditioned on face count, we use the face count of the ground truth mesh as the conditioning input when evaluating reconstruction performance. The generation process is halted when the number of generated faces exceeds twice the specified face count value.

## C.4    TRAINING ON LARGE MESH SEQUENCES BENEFIT SHORT MESH SEQUENCE GENERATION

Previous works, capped at 800 or 1600 faces, are unable to leverage datasets containing larger, more complex meshes during training. This raises a question: can training on longer mesh sequences, even if they exceed the target length, improve the generation of smaller meshes?

To answer this question, we trained a model on data with up to 8K faces –which includes the 4K face training set used for architecture ablations in the main paper–. As shown in Fig. 12, additional training data, even beyond the target sequence length, improves performance on shorter sequence. This finding suggests that training on cropped sequences is a viable strategy for enhancing performance, even when the target is to generate shorter sequences.

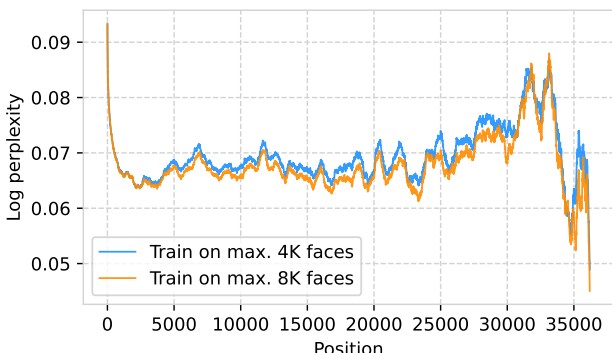

Figure 12: Token position vs. perplexity for models trained on datasets with meshes containing up to 4K faces and up to 8K faces. Both models are evaluated on a validation set with meshes of up to 4K faces (36,864 tokens). Traces are smoothed using a moving average of 576 elements for clarity. The model trained on up to 4K faces achieves a mean perplexity of 1.0671, while the model trained on up to 8K faces achieves a slightly lower mean perplexity of 1.0668 (lower is better).

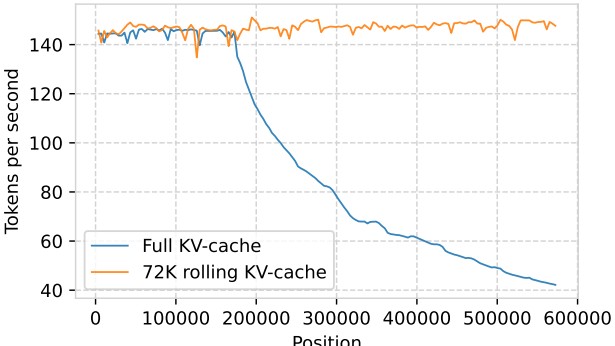

Figure 13: Rolling KV-cache enables linear-time inference. Without rolling KV-cache, inference speed decreases rapidly as the attention span grows. By using rolling KV-cache, MESHTRON maintains a constant generation rate regardless of sequence length.

## C.5 MESHTRON'S ROBUSTNESS OVER POINT CLOUD DENSITIES

We evaluate MESHTRON's robustness over changing point cloud densities. As observed in Fig. 15, MESHTRONdemonstrates strong robustness to variations in the density of the conditioning point clouds. MESHTRONmaintains high-quality generations even when the number of points deviates densities observed during training.

As one could expect, higher point densities provide more detailed geometric information, allowing MESHTRON to generate meshes with finer detail, particularly for complex geometries. Interestingly however, this behavior is observed, even when the specific point cloud density used exceeds those observed during training. Conversely, lower point densities may limit the detail level but still result in coherent and well-structured outputs. This flexibility ensures that MESHTRON can adapt to varying levels of input detail, making it a practical tool for diverse applications where point cloud quality or resolution may vary.

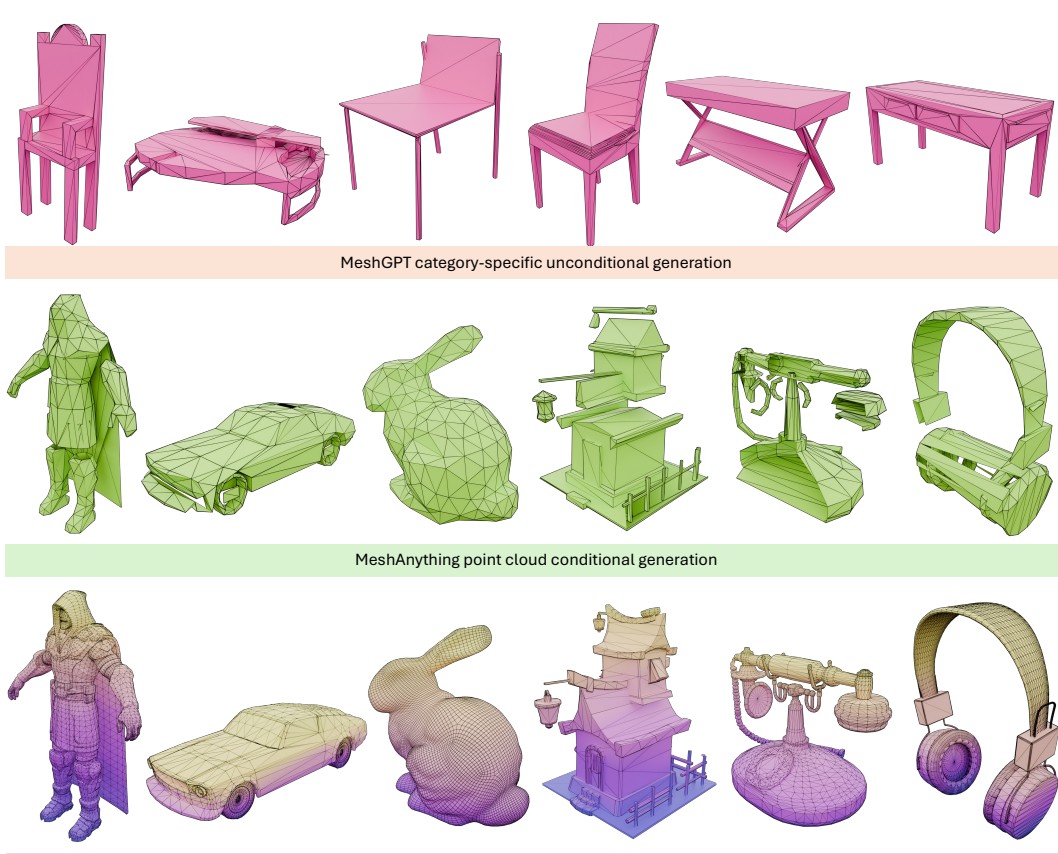

Figure 14: Qualitative comparison of MeshGPT (Siddiqui et al., 2024), MeshAnythingV1 (Chen et al., 2024b) and MESHTRON. MeshGPT and MeshAnything are based on a two-staged model consisting of a pretrained VQ-VAE tokenizer and an autoregressive model. Their generations are capped at a maximum of 800 faces by design. Instead, MESHTRON works directly on the coordinate space and is able to generate meshes up to 64K faces. Please refer to Fig. 8 and Fig. 9 for more comparisons.

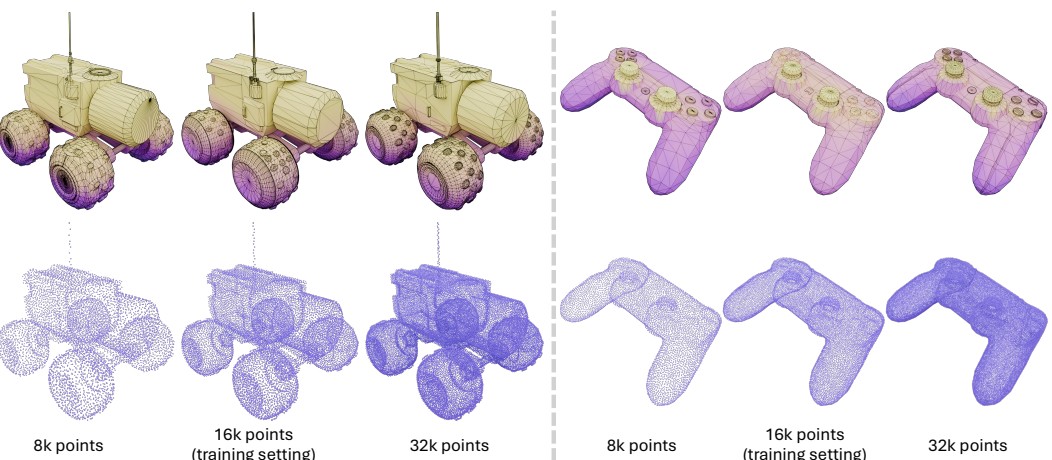

Figure 15: Effect of the number of points on MESHTRON's generations. MESHTRON is robust to variations in the number of points in the conditioning point clouds, even when the specific point count was not observed during training. Interestingly, higher point densities can result in meshes with greater detail –especially for complex geometries– even when using densities exceeding those observed during training.

