# OpenReview forum: "Meshtron: High-Fidelity, Artist-Like 3D Mesh Generation at Scale"
_ICLR.cc/2025/Conference — Submitted to ICLR 2025_

### Official Review · Reviewer_QYFu · 2024-10-27

**Soundness:** 3
**Presentation:** 2
**Contribution:** 3
**Rating:** 5
**Confidence:** 3

**Summary:**

This paper introduces a transformer-based approach to mesh generation. Given an input point cloud, the hourglass architecture (MESHTRON), like an Unet or an AE, compresses and reconstructs the mesh as output. The intermediate states (or latent) have an explicit meaning, corresponding to vertex and face representation. During training the model is trained by exploiting the adjacency of the triangles combined with a sliding window approach. Such a choice reduces memory consumption and allows the model to scale to complex and longer mesh sequences.

**Strengths:**

The proposed approach shows potential. The intermediate latent representation makes use of the geometric construction of meshes. The training procedure allows the architecture to scale to longer and more complex meshes.

**Weaknesses:**

Although the paper is very promising, several details are unclear. I would appreciate it if the authors clarify them:
1. several details are obscured in the manuscript, or not appropriately presented:
   - the meaning of the intermediate representation is not well presented. It can be inferred from Figure 4, but I the authors should to discuss it in the text for the sake of the reader - please expand Section 3.1;
   - it is not clear how the sliding window is used during inference. For example, how is the sliding window decided? And given that the sequence is truncated during training, how is a larger mesh generated? Please, provide more details on the sliding window inference process in Section 3.2.
2. the authors claim the length of the mesh sequence can be 64K in faces, L158-159. However, I cannot find such examples (even in the supplementary material). Is this a theoretical limit or do authors have examples and did not include them in the material?
3. the authors point out "MESHTRON does not work from time to time", would you please elaborate? This is not an appropriate sentence for a scientific paper.
4. how is the global conditioning obtained? Is the input point cloud feed to the model and then the sequence used for cross attention?
5. would it be possible to condition from an image latent representation (say encoded by DinoViT) and project it onto the latent space of MESHTRON? (it is fine if it is future work, but worth mentioning in the paper)
6. a comparison with MeshGPT would be appreciated even if qualitative. As the code is not available, the authors could maybe reach out to MeshGPT's authors to get some examples.

**Questions:**

Please see the weakness section.

---

> ### Author Response · Authors · 2024-11-23
>
> Dear reviewer QYFu,
>
> First of all, we would like to thank you very much for your review. We sincerely appreciate the time you spent in evaluating our work, and very much appreciate your comments.
>
> Here we will answer to your questions, comments and concerns:
>
> **The meaning of the intermediate representation is not well presented. It can be inferred from Figure 4, but I the authors should to discuss it in the text for the sake of the reader - please expand Section 3.1.**
> > We acknowledge that the presentation of the Hourglass architecture could be improved. To address this limitation, we have improved the presentation of the Hourglass architecture in both the Introduction, Sec. 3.1 and Fig. 4 to better highlight its properties and the role of the intermediate representation. In addition, we have included an animation on [**our project website**](https://meshtron.github.io/index.html), which illustrates its inner workings in greater detail. This website will remain active and will be deanonymized upon acceptance.
>
> **it is not clear how the sliding window is used during inference. For example, how is the sliding window decided? And given that the sequence is truncated during training, how is a larger mesh generated? Please, provide more details on the sliding window inference process in Section 3.2.**
> > We have enhanced Sec. 3.2 and Sec. 4.2, adding details to clarify the sliding window inference process. In summary, larger meshes are generated by employing a sliding window (or rolling KV-cache) with a size equal to the training chunk. This ensures consistency between training and inference, eliminating discrepancies caused by sequence truncation. Fig. 7, Sec. 4.2, and Fig. 13 provide further explanation and visualizations to illustrate the importance of this approach. In addition, Appx. A provides additional details regarding how similar approaches have been used by other methods to address similar problems.
>
> **the authors claim the length of the mesh sequence can be 64K in faces, L158-159. However, I cannot find such examples (even in the supplementary material). Is this a theoretical limit or do authors have examples and did not include them in the material?**
> > We have now provided access to several examples on our project website. Note that 64k refers to the upper bound on generations, which is determined by the maximum sequence length used during training. On the project website, you can find examples with over 56K faces, showcasing the model's capability near this limit.
>
> **the authors point out "MESHTRON does not work from time to time", would you please elaborate? This is not an appropriate sentence for a scientific paper.**
> > We apologize for the unscientific phrasing in the original text. We have now replaced it with a precise description of the observed failure modes. Specifically, occasional inference failures manifest as missing parts, holes, distorted (exploding) meshes, or failure to properly terminate the sequence. These issues are rare but highlight the need for further robustness improvements.
>
> **how is the global conditioning obtained? Is the input point cloud feed to the model and then the sequence used for cross attention?**
> > That is correct. The input point cloud is fed to the model and used to generate global embeddings, which are then incorporated through cross-attention layers in the sequence. We have updated Fig. 4 to provide a more comprehensive view of Meshtron's architecture and detailed how the conditioning is achieved both at the beggining of Sec. 4, and in the Appendix.
>
> **would it be possible to condition from an image latent representation (say encoded by DinoViT) and project it onto the latent space of MESHTRON? (it is fine if it is future work, but worth mentioning in the paper)**
> > This is an excellent point. We believe it is indeed possible, given the flexibility of Meshtron's pipeline. The cross-attention architecture, widely used in multimodal LLM research, allows Meshtron to be conditioned on any input modality that can be encoded into one or more embedding vectors. For instance, image latents encoded by DinoViT could be added as additional conditioning during generation. We have highlighted this possibility in the limitations and future work section (Sec. 5).
>
> **a comparison with MeshGPT would be appreciated even if qualitative.**
> > We have included a visual comparison in Fig. 14. The generations observed here are obtained from the official inference code hosted here: https://github.com/audi/MeshGPT . First of all, it is important to highlight that MeshGPT only offers fine-tuned models only able to generate either chairs or tables. Additionally, the quality of the generations is lower than MeshAnything, MeshAnythingV2 and Meshtron.
>
> ---
>
> We hope that these responses clarify your questions and concerns. Please let us know if you have any follow-up / additional questions.
>
> Best regards,
>
> The Authors

---

> > ### Author Response · Authors · 2024-12-02
> > **Reminder: Last opportunity to ask questions to authors.**
> >
> > Dear Reviewer QYFu,
> >
> > We hope this message finds you well.
> >
> > This is a kind reminder that December 2nd at midnight AoE (today) marks the final opportunity to ask authors questions regarding their submissions. In our rebuttal submitted on November 22—10 days ago—we made every effort to address your comments and concerns thoroughly. We believe that all points raised in the review have been addressed in our response and remain eager to address any further questions or clarifications you may have. We would greatly appreciate your engagement during this critical phase of the review process.
> >
> > We value your feedback and hope that our responses have addressed your initial concerns. Please do not hesitate to reach out if there is any additional information or clarification we can provide before the deadline.
> >
> > Thank you for your time and effort in reviewing our work.
> >
> > Best,
> >
> > The authors

---

### Official Review · Reviewer_TBNL · 2024-10-31

**Soundness:** 3
**Presentation:** 2
**Contribution:** 3
**Rating:** 3
**Confidence:** 3

**Summary:**

This paper presents a method for efficiently generating meshes from point clouds. The authors propose an hourglass transformer network architecture and a truncated sequence training scheme, which enable the generation of meshes with a large number of faces and diverse types. Compared to existing methods, this approach requires less training memory and achieves faster throughput.

**Strengths:**

The proposed method addresses an important challenge in 3D mesh generation: the number of faces significantly affects the quality and fidelity of the results, making a higher face count crucial. Experimental results indicate that this method can produce highly detailed meshes.

Additionally, the authors present the motivation for their algorithm design, based on observed issues in real-world data, offering valuable insights for the academic community.

**Weaknesses:**

The author mentions that the tokens of the last vertex have high perplexity, which led to the design of the Hourglass Transformer to address this issue. However, the causal relationship here does not seem entirely clear. Within the architecture of the Hourglass Transformer, there is no obvious special treatment for the last vertex tokens. While the Hourglass Transformer is an effective structure, linking it directly to the perplexity of the last vertex tokens feels somewhat tenuous.


In the experiments, the authors used high-quality point clouds as input. However, in practical applications, point clouds often contain noise. This paper does not discuss or experiment on this issue, which inevitably raises concerns about its practicality.

The author only compares their method with one other approach and does not include a related work section. This makes it difficult for those not working in mesh generation to follow the advancements in the field and compare the proposed method. For instance, how does it differ from existing mesh generation methods such as CLAY[1]?

[1] Zhang L, Wang Z, Zhang Q, et al. CLAY: A Controllable Large-scale Generative Model for Creating High-quality 3D Assets[J]. ACM Transactions on Graphics (TOG), 2024, 43(4): 1-20.

**Questions:**

I believe that the problem this paper aims to address (generating detailed meshes) is meaningful. However, there are many areas for improvement in the experimental design and writing.

---

> ### Author Response · Authors · 2024-11-23
>
> Dear reviewer TBNL,
>
> First of all, we would like to thank you very much for your review. We sincerely appreciate the time you spent in evaluating our work, and very much appreciate your comments.
>
> Here we will answer to your questions, comments and concerns:
>
> **While the Hourglass Transformer is an effective structure, linking it directly to the perplexity of the last vertex tokens feels somewhat tenuous.**
> > We acknowledge that we did adequately explain this aspect in the paper. Hourglass Transformers have two key properties relevant to Meshtron. The first property is that they process outputs in a hierarchical manner. This hierarchy allows the model to effectively process the input at vertex and face levels of abstraction.
>
> > The second property, which contributes to their efficiency but was not properly outlined in the paper, is that Hourglass Transformers dynamically allocate compute to tokens based on their position within the sequence. As a result, some tokens receive more compute than others. This is illustrated in the animation provided in [**our project website**](https://meshtron.github.io/index.html). Specifically, based on the selected shortening --and upsampling-- factors and depending on their position, some tokens will skip the inner Transformer stacks of the Hourglass architecture, whereas others will be processed by the whole architecture. This targeted allocation of compute helps the model focus resources on tokens that require greater attention.
>
> > To make this clearer in the paper, we have: (i) complemented Fig. 5 with an additional per token compute graph that illustrates the per-token compute used for different tokens, and (ii) improved the explanations provided in the Introduction as well as Sec. 3.1. In addition, we will keep the animation on the project website, which we will de-anonymize upon acceptance.
>
> > We hope this clarifies this connection and are happy to complement this with any other observations the reviewer might have.
>
> **In the experiments, the authors used high-quality point clouds as input. However, in practical applications, point clouds often contain noise. This paper does not discuss or experiment on this issue, which inevitably raises concerns about its practicality.**
> > This is a good observation. We originally provide some brief information regarding this in the Appendix, but it was not mentioned in the main text. During training, we augment the point cloud positions and normals with gaussian noise to simulate noisy real-world conditions. In addition, we make sure that point clouds are never sampled from internal structures --which may be present in artist meshes, but are unlikely to appear in scanned or automatically generated meshes-- to aid generalizability to non-artist like meshes. This treatment is crucial, and it is in fact, thanks to this that Meshtron is also able to receive point-clouds from scans as well as iso-surfacing methods (which are often noise and imperfect) and successfully map them to artist-quality meshes.
>
> > To address your concern, we have updated Sec. 4 to include details regarding this treatment and their role in enhancing robustness. Additionally, we have added Fig. 11, which illustrates how adjusting input noise allows Meshtron to balance creativity and faithfulness, showcasing its flexibility and applicability in scenarios with varying levels of input quality.

---

> > ### Author Response · Authors · 2024-11-23
> >
> > **The author only compares their method with one other approach and does not include a related work section. This makes it difficult for those not working in mesh generation to follow the advancements in the field and compare the proposed method. For instance, how does it differ from existing mesh generation methods such as CLAY[1]?**
> >
> > > We appreciate this feedback and agree that broader validation strengthens our claims. In response, we have added comparisons to iso-surfacing methods FlexiCubes and DMTet, as well as native mesh methods MeshGPT and MeshAnythingV1. Unfortunately, we could not find inference code for CLAY online. However, CLAY also makes use of iso-surfacing methods to obtain Meshes from alternative 3D representations, therefore making our analysis on iso-surfacing methods also apply to it. These comparisons are detailed in the Introduction and Sec. 4.3. Visual comparisons are provided in Fig 2 and Fig 14, in addition to a very large gallery of results and additional comparisons in our project website.
> >
> > > In summary, in comparison to methods that generate meshes directly (MeshGPT, MeshAnythingV1 / V2), Meshtron is able to generate meshes with many more faces at higher resolution, giving it the ability to create artist-like meshes of complex 3D objects. When compared to iso-surfacing methods (DMTet, FlexiCubes, CLAY), which can produce meshes with very high face counts, Meshtron is able to do so while obtaining high-quality topology, featuring high-geometric detail and well-structured tessellation, just at it is observed in artist-like meshes. Iso-surfacing methods, on the other hand, often suffer from overly dense tessellation, bumpy artifacts, oversmoothing and insufficient geometric detail, creating a notable quality gap between their meshes and those crafted by artists.
> >
> > > Additionally, we agree that a dedicated related work section is essential for helping readers contextualize the advancements in the field and appreciate the value of Meshtron. We have now added an extended related work section to our manuscript.
> >
> > ---
> >
> > We hope that these responses clarify your questions and concerns. Please let us know if you have any follow-up / additional questions.
> >
> > Best regards,
> >
> > The Authors

---

> > > ### Author Response · Authors · 2024-12-02
> > > **Reminder: Last opportunity to ask questions to authors.**
> > >
> > > Dear Reviewer TBNL,
> > >
> > > We hope this message finds you well.
> > >
> > > This is a kind reminder that December 2nd at midnight AoE (today) marks the final opportunity to ask authors questions regarding their submissions. In our rebuttal submitted on November 22—10 days ago—we made every effort to address your comments and concerns thoroughly. We believe that all points raised in the review have been addressed in our response and remain eager to address any further questions or clarifications you may have. We would greatly appreciate your engagement during this critical phase of the review process.
> > >
> > > We value your feedback and hope that our responses have addressed your initial concerns. Please do not hesitate to reach out if there is any additional information or clarification we can provide before the deadline.
> > >
> > > Thank you for your time and effort in reviewing our work.
> > >
> > > Best,
> > >
> > > The authors

---

### Official Review · Reviewer_rUX9 · 2024-11-03

**Soundness:** 3
**Presentation:** 3
**Contribution:** 3
**Rating:** 5
**Confidence:** 2

**Summary:**

The author presents a methodology for generating high-quality mesh generation from point clouds. The author proposes an autoregressive model that has been well evaluated and chosen regarding the explicit need for mesh generation. The authors claim well-analysed contributions in the chosen neural architecture and the training methodology and reconstruct the mesh by limiting the scope of possible solutions to enforce mesh sequences during training and inference.

**Strengths:**

The work is well ablated, and the choice in architectural design taken by the authors is reasonable and easy to understand. Overall, the work is well written and well structured, making it easy to understand why and what was chosen and how it was implemented. The appendix is additionally very useful to reimplement this work. Overall, the work seems to have well-reasoned analysis for training mesh generations from autoregressive models and additionally adds limitations to the mesh generation process that seemingly seem to be very impactful in this process.

**Weaknesses:**

I am not an expert in this research area, but for me, the main weakness is the claim that previous work "recent attempts at generating artist-like meshes are limited to 1.6K faces and heavy discretization of vertex coordinates". I want to refer the authors to works such as [1, 2] that show mesh reconstruction with more that 1.6K faces.
Both of the mentioned works [1, 2] show high-quality reconstructions and are not compared by the authors.

[1] Shen, Tianchang, et al. "Flexible Isosurface Extraction for Gradient-Based Mesh Optimization." ACM Trans. Graph. 42.4 (2023): 37-1.
[2]Shen, Tianchang, et al. "Deep marching tetrahedra: a hybrid representation for high-resolution 3d shape synthesis." Advances in Neural Information Processing Systems 34 (2021): 6087-6101.

**Questions:**

- How much does the number of points in the point cloud impact the final mesh generation process
- How does the model handle partial point clouds?
- Above all, I would like the point regarding the comparision with [1, 2] to be addressed and would change my rating if it was a misunderstanding on my side.


[1] Shen, Tianchang, et al. "Flexible Isosurface Extraction for Gradient-Based Mesh Optimization." ACM Trans. Graph. 42.4 (2023): 37-1.
[2]Shen, Tianchang, et al. "Deep marching tetrahedra: a hybrid representation for high-resolution 3d shape synthesis." Advances in Neural Information Processing Systems 34 (2021): 6087-6101.

---

> ### Author Response · Authors · 2024-11-23
>
> Dear reviewer rUX9,
>
> First of all, we would like to thank you very much for your review. We sincerely appreciate the time you spent in evaluating our work, and very much appreciate your comments.
>
> Here we will answer to your questions, comments and concerns:
>
> **The main weakness is the claim that previous work "recent attempts at generating artist-like meshes are limited to 1.6K faces and heavy discretization of vertex coordinates". I want to refer the authors to works such as [1, 2] that show mesh reconstruction with more that 1.6K faces. Both of the mentioned works [1, 2] show high-quality reconstructions and are not compared by the authors.**
>
> > We acknowledge that we did not sufficiently clarify the problem we are addressing and how other methods, despite generating meshes with large face counts, fall short of solving it. To address this, we have revised the introduction and included comparisons with the iso-surfacing methods DMTet [2] and FlexiCubes [1] in Fig. 2.
>
> > In a nutshell, the key limitation of iso-surfacing methods is that they often exhibit poor topology, characterized by overly dense tesselation, over-smoothing, and bumpy artifacts, which results in a significant quality gap between AI-generated meshes and those crafted by artists (see Fig. 2). Consequently, the topology of these methods is insufficient for practical downstream applications.
>
> > To overcome this limitations, methods such as MeshGPT and MeshAnythingV1/V2  attempt to model 3D assets directly as meshes. While the topology of these methods is much closer to those created by artists, these methods are heavily constrained by the computational cost of processing the (very) long sequences required for high face counts. Meshtron provides an scalable alternative to generate artist-like meshes composed of many faces --40x more than existing works--, comparable to the face count of iso-surfacing methods (DMTet, FlexiCubes), but providing high-quality topology that resembles that of meshs crafted by artsits. Comparisons and additional generations have been added both to the paper as well as [**our project website**](https://meshtron.github.io/index.html).
>
> **How much does the number of points in the point cloud impact the final mesh generation process**
> > We have complemented our manuscript with an additional analysis of this aspect. This effect is shown in Fig 15 and Appx. C5. In a nutshell, we observe that  Meshtron high-quality generations even when the number of points deviates from densities observed during training. As one could expect, higher point densities provide more detailed geometric information, allowing Meshtron to generate meshes with finer detail, particularly for complex geometries, even when the specific point cloud density used exceeds those
> observed during training. Conversely, lower point densities may limit the detail level but still result in coherent and well-structured outputs.
>
> **How does the model handle partial point clouds?**
> > We did not consider this application in this project. However, we envision that Meshtron could either be used in tandem with point cloud completion methods to progressively generate high-quality meshes from incomplete point clouds, or be trained with point cloud corruptions to enable the model itself to handle partial point clouds. Both approaches could be promising directions for future work.
>
> **Above all, I would like the point regarding the comparision with [1, 2] to be addressed and would change my rating if it was a misunderstanding on my side.**
> > We hope that our previous response, along with the revisions and additional comparisons provided, have addressed this concern. Please do not hesitate to let us know if further clarifications or additional information would be helpful.
>
> ---
>
> We hope that these responses clarify your questions and concerns. Please let us know if you have any follow-up / additional questions.
>
> Best regards,
>
> The Authors

---

> > ### Author Response · Authors · 2024-12-02
> > **Reminder: Last opportunity to ask questions to authors.**
> >
> > Dear Reviewer rUX9,
> >
> > We hope this message finds you well.
> >
> > This is a kind reminder that December 2nd at midnight AoE (today) marks the final opportunity to ask authors questions regarding their submissions. In our rebuttal submitted on November 22—10 days ago—we made every effort to address your comments and concerns thoroughly. We believe that all points raised in the review have been addressed in our response and remain eager to address any further questions or clarifications you may have. We would greatly appreciate your engagement during this critical phase of the review process.
> >
> > We value your feedback and hope that our responses have addressed your initial concerns. Please do not hesitate to reach out if there is any additional information or clarification we can provide before the deadline.
> >
> > Thank you for your time and effort in reviewing our work.
> >
> > Best,
> >
> > The authors

---

### Official Review · Reviewer_hX86 · 2024-11-16

**Soundness:** 3
**Presentation:** 2
**Contribution:** 3
**Rating:** 5
**Confidence:** 3

**Summary:**

The authors propose an autoregressive approach for generating high-quality meshes from input point clouds. Compared to existing methods, this approach achieves over an order of magnitude increase in face count. The introduction of the Hourglass Transformer and truncated sequence training effectively reduces training memory requirements and enhances computational efficiency.

**Strengths:**

The paper addresses the challenge of 3D mesh generation by achieving high-quality outputs with up to 64K faces at 1024-level coordinate resolution, representing a significant advancement over existing methods. The intuition and motivation behind the approach are clearly articulated, supported by a thoughtful discussion of real-world data. The authors further validate their method by conditioning on a variety of mesh types, including artist-created meshes and text-to-3D generated meshes, demonstrating its versatility.

**Weaknesses:**

First, I find it unclear why the Hourglass Transformer is considered the appropriate solution to address the challenge of generating the latter tokens of a triangle. While the authors explain both the difficulty and the Hourglass Transformer reasonably well, the connection between the two could be more explicitly justified. Second, the model’s reliance on curated, high-quality datasets raises concerns about scalability and applicability to domains with limited or noisy data. In real-world scenarios, point clouds are often noisy, yet the paper does not evaluate the robustness of the proposed approach under such conditions. Finally, although the authors claim significant advantages over state-of-the-art methods, they only compare their approach to one such method. Broader validation and comparisons with a wider range of methods are necessary to substantiate their claims.

**Questions:**

While the 64K-face mesh generation represents a notable improvement over existing methods, I question whether this represents the theoretical limit of the proposed approach. In real-world scenarios involving complex scenes, mesh face counts often exceed several million, raising concerns about the scalability of the method to handle such high levels of complexity.

---

> ### Author Response · Authors · 2024-11-23
>
> Dear reviewer hX86,
>
> First of all, we would like to thank you very much for your review. We sincerely appreciate the time you spent in evaluating our work, and very much appreciate your comments.
>
> Here we will answer to your questions, comments and concerns:
>
> **First, I find it unclear why the Hourglass Transformer is considered the appropriate solution to address the challenge of generating the latter tokens of a triangle.**
>
> > We acknowledge that we did adequately explain this aspect in the paper. Hourglass Transformers have two key properties relevant to Meshtron. The first property is that they process outputs in a hierarchical manner. This hierarchy allows the model to effectively process the input at vertex and face levels of abstraction.
>
> >  The second property, which contributes to their efficiency but was not properly outlined in the paper, is that Hourglass Transformers dynamically allocate compute to tokens based on their position within the sequence. As a result, some tokens receive more compute than others. This is illustrated in the animation provided in [**our project website**](https://meshtron.github.io/index.html). Specifically, based on the selected shortening --and upsampling-- factors and depending on their position, some tokens will skip the inner Transformer stacks of the Hourglass architecture, whereas others will be processed by the whole architecture.  This targeted allocation of compute helps the model focus resources on tokens that require greater attention.
>
> > To make this clearer in the paper, we have: (i) complemented Fig. 5 with an additional per token compute graph that illustrates the per-token compute used for different tokens, and (ii) improved the explanations provided in the Introduction as well as Sec. 3.1. In addition, we will keep the animation on the project website, which we will de-anonymize upon acceptance.
>
> **Second, the model’s reliance on curated, high-quality datasets raises concerns about scalability and applicability to domains with limited or noisy data. In real-world scenarios, point clouds are often noisy, yet the paper does not evaluate the robustness of the proposed approach under such conditions.**
> > This is a good observation. We originally provide some brief information regarding this in the Appendix, but it was not mentioned in the main text. During training, we augment the point cloud positions and normals with gaussian noise to simulate noisy real-world conditions. In addition, we make sure that point clouds are never sampled from internal structures --which may be present in artist meshes, but are unlikely to appear in scanned or automatically generated meshes-- to aid generalizability to non-artist like meshes. This treatment is crucial, and it is in fact, thanks to this that Meshtron is also able to receive point-clouds from scans as well as iso-surfacing methods (which are often noise and imperfect) and successfully map them to artist-quality meshes.
>
> > To address your concern, we have updated Sec. 4 to include details regarding this treatment and their role in enhancing robustness. Additionally, we have added Fig. 11, which illustrates how adjusting input noise allows Meshtron to balance creativity and faithfulness, showcasing its flexibility and applicability in scenarios with varying levels of input quality.
>
> **Finally, although the authors claim significant advantages over state-of-the-art methods, they only compare their approach to one such method. Broader validation and comparisons with a wider range of methods are necessary to substantiate their claims.**
> > We appreciate this feedback and agree that broader validation strengthens our claims. In response, we have added comparisons to iso-surfacing methods FlexiCubes and DMTet, as well as native mesh methods MeshGPT and MeshAnythingV1. These comparisons are detailed in the Introduction and Sec. 4.3. Visual comparisons are provided in Fig 2 and Fig 14, in addition to a very large gallery of results and additional comparisons in our project website.
>
> > In summary, in comparison to methods that generate meshes directly (MeshGPT, MeshAnythingV1 / V2), Meshtron is able to generate meshes with many more faces at higher resolution, giving it the ability to create artist-like meshes of complex 3D objects. When compared to iso-surfacing methods (DMTet, FlexiCubes), which can produce meshes with very high face counts, Meshtron is able to do so while obtaining high-quality topology, featuring high-geometric detail and well-structured tessellation, just at it is observed in artist-like meshes. Iso-surfacing methods, on the other hand, often suffer from overly dense tessellation, bumpy artifacts, oversmoothing and insufficient geometric detail, creating a notable quality gap between their meshes and those crafted by artists.
>
> —
>
> We hope that these responses clarify your questions and concerns. Please let us know if you have any follow-up / additional questions.
>
> Best regards,
>
> The Authors

---

> > ### Author Response · Authors · 2024-12-02
> > **Reminder: Last opportunity to ask questions to authors.**
> >
> > Dear Reviewer hX86,
> >
> > We hope this message finds you well.
> >
> > This is a kind reminder that December 2nd at midnight AoE (today) marks the final opportunity to ask authors questions regarding their submissions. In our rebuttal submitted on November 22—10 days ago—we made every effort to address your comments and concerns thoroughly. We believe that all points raised in the review have been addressed in our response and remain eager to address any further questions or clarifications you may have. We would greatly appreciate your engagement during this critical phase of the review process.
> >
> > We value your feedback and hope that our responses have addressed your initial concerns. Please do not hesitate to reach out if there is any additional information or clarification we can provide before the deadline.
> >
> > Thank you for your time and effort in reviewing our work.
> >
> > Best,
> >
> > The authors

---

### Meta-Review · Area_Chair_c43c · 2024-12-14

**Metareview:**

The paper proposes an autoregressive method to generate meshes from point clouds. The method is claimed to produce meshed of up to 64K faces at higher resolution than prior work and resembling meshes created by artists in terms of detail and quality. Technical contributions are an hourglass network architecture, a training protocol, a sliding-window based inference procedure, and a robust sampling strategy.

The reviewers found the paper well motivated, the technical contributions well described, and the architecture design choices well explained.
The paper provides quantitative results comparing architecture variants and for training on truncated sequences and qualitative comparisons to prior work.

Concerns and requests raised during the discussion period were sufficiently addressed to a large extent. The main weakness of the paper however, is the limited comparison to prior work. Initially, a related work section was missing completely, but was added to the appendix in the revised version after requests were made by the reviewers. The requests for additional experimental validation and comparisons to prior work was only met through qualitative results. Although the qualitative results indeed show improvements over prior work, a more thorough, quantitative comparison on a larger set of samples would be necessary to sufficiently validate the proposed method (Even the revised version does not contain a single quantitative comparison against prior work.)

**Additional Comments On Reviewer Discussion:**

Discussion:
- Limited evaluation [hX86,TBNL] was addressed through few qualitative examples
- Missing comparison to prior work [rUX9,TBNL] was addressed through qualitative comparison.
- Reliance on high quality dataset, unclear performance on out of domain data and robustness to noise [hX86,TBNL]  was addressed through few qualitative examples

- Justification of hourglass transformer [hX86] was given
- Missing related work section [TBNL] was handled partially through intro, adressed through rebuttal.
- Insufficient link between last token perplexity and architecture [TBNL] adressed in revised version
- Insufficient architectural details (sliding window attention, conditioning on point cloud) [QYFu] addressed in revision
- Example for 64K sequence as claimed, theoretical limit [hX86,QYFu] was addressed through website
- Failure during inference quote [QYFu] addressed through revision

Several concerns raised by the reviewers were addressed through added details and animation on the project website.
The AC does not consider this an adequate means of delivering results as it allows updating results beyond the end of the discussion period,
does not record any modifications, circumvents formatting constraints that other papers also adhere to, and poses a (limited) risk to the anonymity of the reviewers through tracking.

---

### Decision · Program_Chairs · 2025-01-22

Reject